# Explanatory Journalism within European Fact Checking Platforms: An Ally against Disinformation in the Post-COVID-19 Era

Victoria Moreno-Gil [1,*], Xavier Ramon-Vegas [2,*], Ruth Rodríguez-Martínez [2] and Marcel Mauri-Ríos [2]

1  Communication and Media Studies Department, Carlos III University of Madrid, 28903 Madrid, Spain
2  Department of Communication, Pompeu Fabra University, 08018 Barcelona, Spain;
   ruth.rodriguez@upf.edu (R.R.-M.); marcel.mauri@upf.edu (M.M.-R.)
*  Correspondence: vicmoren@hum.uc3m.es (V.M.-G.); xavier.ramon@upf.edu (X.R.-V.)

**Abstract:** In the post-COVID era, explanatory journalism is undergoing a resurgence that can be attributed to the proliferation of false content disseminated via social networks and the maturation of fact checking initiatives. Fact checkers are beginning to delve into those topics that are recurrent targets of disinformation to make complex issues accessible to the public. This study investigates the characteristics and methodologies of contemporary explanatory journalism by analysing four European verification platforms (Newtral in Spain, Les Décodeurs in France, FACTA.news in Italy and The Journal FactCheck unit in Ireland). We employed content analysis of a corpus of explainers and semi-structured interviews with the managers of these outlets. Our findings reveal that explainers encompass a wide range of topics, typically revolving around current affairs. These pieces are usually authored by fact checkers and published, with bylines, within dedicated sections that encourage audience participation. Explainers do not adhere to a fixed periodicity or length and adopt a format similar to feature articles, displaying a degree of flexibility. They leverage data provided by experts and official sources and employ visual elements to convey information clearly. The interviewed managers concur that explanatory journalism represents an invaluable tool in combatting disinformation and has a promising future ahead.

**Keywords:** explanatory; journalism; fact checking; Europe; disinformation; post-COVID

## 1. Introduction

Recent research into verification and disinformation underscores the strong relationship between journalism and fact checking [1–3]. This connection is explained not only by the fact that most members of fact checking teams tend to have journalistic backgrounds or prior media experience, or that they embody what has come to be called new professional profiles—including data analysts, visual journalists or infographic specialists—but is also due to the very nature of the fact checking methodology. This methodology has evolved into an enhanced iteration of traditional source consultation, "an indispensable dimension, shared by all journalism professionals throughout its modern history" [4] (p. 1). Some scholars even argue that fact checking embodies a quasi-scientific [5] form of objectivity [6] that surpasses the declarative style of journalism —a contemporary classic in the field.

For other authors, modern fact checking is not merely a practice but represents an "emergent form of accountability reporting" [1] (p. 624) and "journalistic genre" [7] (p. 2). Many fact checking initiatives have originated from traditional media outlets [8], further underscoring the close-knit relationship between journalism and fact checking.

As Singer suggests [9] (p. 1930), a key to understanding the role of modern fact checkers within the sphere of journalism is their approach to the "fundamental norm of truth-telling". While journalists tend to prioritise accurately reporting what was said, fact

checkers are more concerned with evaluating the accuracy of statements. Likewise, the social function of journalism serves as a guiding principle for the work of fact checkers, who are reviving the profession's core values [10], harking back to its origins [5]. Within this context, explanatory texts have recently emerged as common ground between the two fields.

Explanatory journalism is not a new phenomenon, nor is it solely linked to modern fact checking. Its presence has steadily grown among leading U.S. newspapers since the 1950s [11]. Its finest works were honoured with the Pulitzer Prize for Explanatory Journalism until 1997 and, subsequently, with the Pulitzer Prize for Explanatory Reporting from 1998 onwards. However, in recent years, especially with the rise of data journalism and fact checking organisations, explanatory journalism has experienced exponential growth [1,9,12–14].

Svith [15] defines explanatory journalism as a growing practice that transcends conventional journalism, adding a new dimension that highlights the "how" and "why" while providing context and additional information on the significance of the phenomenon. This aspect has gained particular prominence since the onset of the coronavirus pandemic, which exacerbated the issue of disinformation through an unprecedented proliferation of hoaxes [16].

For other authors, explanatory journalism represents a departure from the immediacy and superficiality of breaking news. Instead, it aims to elucidate complex subjects in a clear and accessible manner for the general public [12,17]. As Dan and Rauter [13] (p. 1047) point out, "compared with conventional news, explanatory reporting facilitates a superior understanding of the issue at hand, exhibits a greater potential to alter attitudes, and helps clarify the role of macro-level factors in generating and resolving problems". The noteworthy contribution of explanatory journalism in shaping the public's understanding of events is crucial: "news organisations able to invest in explanatory journalism stand better chances of contributing to an informed citizenry than those focussed solely on conventional reporting" [13] (p. 1062).

In essence, it is a traditional form of reporting that the profession has naturally gravitated towards, resulting in progressively longer and more interpretative texts [18]. This extended length is not solely intended to provide context but also to accommodate "more sources and the inclusion of more points of view than feasible in conventional reporting" [13] (p. 1049).

Some experts use the term contextual journalism, which they define as "powerful and prevalent" [19] (p. 1658) and as a frequent companion to traditional reporting. Its impact on how people comprehend their world has yet to be analysed [11]. They highlight that contextual reporting, unlike investigative journalism, may not necessarily involve interactions with confidential sources in the style of Watergate but instead seeks to uncover existing information that has not yet been examined [11]. The contextual journalism genre encompasses several specific reporting styles, including constructive journalism [20,21], solutions journalism or restorative narrative journalism [19,22].

## 2. Explanatory Journalism in the Disinformation Era

With the emergence of the first fact checking organisations in the early 2000s in the United States, a relationship began to develop between fact checking, media literacy and accountability [23]. Initially, fact checkers primarily focused on countering false political discourses [1,10,24]. However, everything changed with the ascent to power of populists such as Donald Trump (2017–2021 in the United States) and Jair Bolsonaro (2019–2023 in Brazil) [25,26] and, more recently, with the COVID-19 crisis.

The pandemic thrust science and health-related topics into a new spotlight as a result of a "surge of hoaxes", giving rise to an "infodemic" [16] (p. 2). Additionally, as noted by Waller and Brookes [27] (p. 45), "the emergence of the COVID-19 pandemic at a time where the impacts of information disorder were already sending shockwaves through the global

information landscape, has created a 'disruptive moment' where media power is being contested by a range of social actors throughout the media world".

In the face of the current onslaught of disinformation, which spreads at alarming speed [28], fact checking platforms, aware of the significant risks this poses to democracies—including increasing polarisation [29]—have begun to take action against the viral proliferation of hoaxes and fake news [30]. These platforms also focus on contentious and frequently recurring content, whose credibility is questioned by citizens [31].

Hence, the resurgence of false content going viral on the internet and social networks has led to the growth of explanatory journalism. Fact checkers have begun to adopt this format to explore "topics that are frequently targeted by disinformation" in a bid "to provide substantiated answers on, for example, election days, vote counting or the functioning of public administrations" [31] (p. 245).

Two other key factors can explain the rise of explanatory journalism. Firstly, the maturation of fact checking projects, which, more than a decade after their inception and despite their ongoing challenges, have succeeded in establishing robust and transparent working methodologies. This is evident in recent studies on the phenomenon in Spain [32,33]; in Spain and Italy [3]; in Spain, France, Italy and Portugal [5]; in Spain and Latin America [34,35] and in Latin America [36].

At the same time, these organisations have created channels for public participation hitherto unknown, acknowledging their pivotal role in rebuilding public trust [37]. Through these avenues, verifiers receive both fact checking requests and queries about specific topics. This type of participatory journalism fosters a close and, at times, mutually dependent relationship between the fact checker and the reader, to which verification organisations respond by providing more media literacy tools and additional contextual information in their pieces.

The second factor that demands attention is the crisis faced by the traditional journalism model. Ryfe [18] states that when the profession loses its monopoly over the traditional gatekeeper function in favour of multiple voices [38], its primary objective shifts from merely being the first to report the news to providing in-depth explanations.

For all the above reasons, Bielik and Višňovský argue [12] (p. 30) that the profession "must therefore devote its primary resources to explanatory journalism, which deals in depth with political issues, trials, institutions in a democratic society and, through thorough explanation, also questions myths and concerns about surrounding issues such as immigration, terrorism and other issues that perpetuate the audience's attention".

The U.S. organisation Politifact published its first explanatory article in August 2007, the same year it became affiliated with the Florida newspaper *Tampa Bay Times*. In 2018, the daily was acquired by the Poynter Institute, which transformed Politifact's business model from that of a "newsroom" to that of an "NGO", as defined by Graves and Cherubini [8]. In Latin America, Chequeado, a prominent fact checking project, launched a section called "El Explicador" in 2010, the same year the platform was set up.

It is mainly due to the coronavirus pandemic that explanatory journalism gained greater visibility within Spanish fact checking platforms, although the first initiative explicitly linked to the explanatory format emerged earlier, specifically with the launch of "Maldita Explica" in 2018. In that section, articles were published that delved into topics that raised doubts among citizens and were susceptible to disinformation.

Despite an increasing number of fact checking outlets dedicating effort and resources to the creation of explainers, the phenomenon of explanatory journalism as part of the work of fact checkers, along with its defining features and methodology, has yet to be addressed in the academic literature. Based on the aforementioned, several key questions emerge that we attempt to address throughout this research, including: When and why do fact checkers produce explanatory articles as part of their daily work? What are the formal and content characteristics of explainers, and what methodology do fact checkers follow when publishing these pieces? How does explanatory journalism contribute to the fight against false information in the post-pandemic era?

### 3. Objectives and Methodology

Rodríguez-Pérez [30] has focused on fact checking journalism initiatives that solely verify fraudulent content and those that address dubious content while aiming to promote media literacy and explanatory journalism. This article focuses on the second type of initiative, with its objective being to explore the key to the explanatory journalism phenomenon, the causes of its origin and its relevance in countering disinformation in the post-COVID era within European fact checking platforms (Table 1). To achieve this, we analysed four private fact checking organisations (Les Décodeurs in France, Newtral in Spain, FACTA.news in Italy and The Journal FactCheck unit in Ireland), with the last three being signatories of the Code of Principles of the International Fact Checking Network (IFCN). It is worth noting that Les Décodeurs obtained this status in 2017, although it currently appears as expired on the IFCN's website (since 20 August 2021).

**Table 1.** Fact checking platforms analysed by country, year of establishment, explainer section and year of launch of the section.

|  | Country | Year of Creation | Year of Launch Explanatory Section | Explanatory Section |
|---|---|---|---|---|
| Newtral | Spain | 2018 | 2018 and 2019 | "You ask us about" and "We explain it to you" |
| Les Décodeurs | France | 2019 | 2020 | "Pour comprendre" |
| FACTA.news | Italy | 2020 | 2021/2022 * | "Articoli" and "Storie" |
| The Journal FactCheck unit | Ireland | 2016 | 2017 | "FactFind" and "Explainer" |

Source: Own elaboration. * The section was launched on 31 December 2021.

The selection of these platforms is motivated by the need to explore the phenomenon in different geographical contexts across Europe and based on different working models. FACTA.news corresponds to the NGO model, while Les Décodeurs, affiliated with the newspaper *Le Monde*, and The Journal FactCheck unit, a verification project linked to *The Journal*, belong to the so-called newsroom model [8]. On the other hand, Newtral falls within for-profit organisations, as it began as a startup that, in addition to fact checking, engages in other activities in audiovisual production. The four organisations have pioneered the introduction and consolidation of the use of explanatory articles within their fact checking propositions. As with other qualitative research, the study endeavoured to understand the practices, methodologies and strategies applied by those organisations rather than aiming to generalise the findings to all the populations [39], that is, to all the fact checking platforms operating in the four countries.

The research questions guiding the study are as follows:

RQ1. What are the defining features of explanatory articles published on the fact checking platforms under study: volume of publication and periodicity, main topics, inclusion of the journalist bylines, breakdown of sources consulted, incorporation of graphic and/or multimedia elements to accompany the explainers, and channels open to audience participation?

RQ2. What factors explain the rise of explanatory journalism within fact checking organisations?

RQ3. What is the methodology employed in creating explainers, and how does it differ from the approach used in traditional fact checking?

RQ4. How does explanatory journalism contribute to combatting disinformation in the post-COVID era? Can this approach be considered an established trend in the field of fact checking?

To respond to these questions, we employed a mixed-method approach. Bringing together quantitative and qualitative techniques allowed researchers to "triangulate findings in order to mutually corroborate them" [39] (p. 557) while helping them to obtain a fuller account of the phenomena under study. First, explanatory texts published on the selected platforms during 2022 were examined to collate the pieces' distinctive char-

acteristics through content analysis, a technique previously used in various studies on disinformation [40,41]. The analysis encompassed seven categories:

1.  Frequency of publication of explanatory pieces.
2.  Number of explainers published.
3.  Overriding topics.
4.  Presence or absence of the journalist's byline in the articles.
5.  Breakdown of consulted sources.
6.  Usage and type of visual elements and/or multimedia resources.
7.  Channels open to audience participation.

The study was complemented by the results of semi-structured interviews with the managers of the four outlets under study: Joaquín Ortega (Newtral), Jonathan Parienté (Les Décodeurs), Giovanni Zagni (FACTA.news) and Shane Raymond (The Journal FactCheck unit). This methodological approach has been widely used in recent research on the fact checking phenomenon [3,5,7,32–34,42–44].

The questions posed to the interviewees (see Appendix A) revolved around the following points: When did they start publishing explanatory texts, and how did the need for them arise? What are the most frequently covered topics, and to what extent does the COVID-19 pandemic explain the rise of explanatory journalism? How often do they publish these pieces? What methodologies do they use in their preparation (consulted sources and pre-publication filters, and do these differ from those applied in traditional fact checks? Who typically writes the texts, and do they usually include their bylines)? What target public are the explanatory articles aimed at, and what feedback and audience participation does the platform receive?

Additionally, the interviewees were asked to provide their perspectives on how explanatory journalism can help to combat disinformation and, finally, whether this type of journalism can be considered a new trend within the fact checking sphere. The interviews were conducted via videoconference between November 2022 and March 2023 and were recorded for subsequent transcription and qualitative analysis.

## 4. Results

Drawing upon the materials gleaned from the content analysis and semi-structured interviews with platform managers (Newtral, Les Décodeurs, FACTA.news and The Journal FactCheck unit), the following results (Table 2) are structured around the three research questions. Firstly, we address the characteristics of the explainers published by these platforms in 2022. Then we shift our focus to the factors that elucidate their commitment to explanatory journalism and the methodologies employed by these organisations in their work. The section concludes by examining the perceptions of the interviewed fact checkers regarding the value of the explanatory format in countering disinformation.

### 4.1. Characteristics of Explanatory Articles (RQ1)

In 2022, Newtral published 149 explainers within the section "You ask us about", averaging three per week, or approximately 12.41 per month. These explainers covered a diverse range of topics, with the most frequently addressed subjects, in descending order, being Russia's invasion of Ukraine, COVID-19, disinformation related to Catalan separatism and associated policies, the LGTBI collective, the transgender law, the "only yes means yes" law, as well as contentious content rooted in xenophobic narratives and anti-immigration discourses. Additionally, climate change was a topic of significant focus. All these explanatory pieces are attributed to specific authors. However, in 19 instances where Newtral collaborated with Verificat, the explainers were instead attributed to "Newtral and Verificat". In four cases, the pieces bore the signature "Newtral", while one article was attributed to "Newtral Data". Notably, 15 explainers were the result of collaborative efforts involving two team members, and in some cases, up to five journalists were credited with the content creation process.

**Table 2.** Summary of the results of the analysis of explainers published on the selected fact checking platforms in 2022.

| | Number of Texts | Weekly Frequency | Topics Covered | Byline | Sources | Visual Elements | Participation |
|---|---|---|---|---|---|---|---|
| Newtral | 149 | 3 | Russia's invasion of Ukraine; COVID-19; Catalan separatism; LGBTI; transgender and "only yes means yes" laws; xenophobia; climate change | Yes | Yes | Yes (screenshots of tweets, images, graphics and videos) | Yes |
| Les Décodeurs | 48 | 1 | National politics; international politics; Russia's invasion of Ukraine; climate change; economy; history; education; justice; COVID-19; vaccines | Yes | Yes | Yes (graphics, videos, infographics, maps and statistics) | No |
| FACTA.news | 82 (55 + 27) | 1.4 * | COVID-19; Russia's invasion of Ukraine; LGBTI; science; climate change; national politics; sports; international politics; social media; xenophobia | No | Yes | Yes (screenshots of tweets, graphics, videos, images and maps) | Yes |
| The Journal FactCheck unit | 9 | 1 ** | Climate change; national politics; education; science; education; transport; tourism | Yes | Yes | Yes (screenshots of tweets, images, tables and maps) | Yes |

Source: Own elaboration. * The figure provided by Giovanni Zagni (director of FACTA.news) takes into account the current publication frequency, which was lower in early 2022 but increased to more than one article per week by late 2022. ** The figure provided by Shane Raymond (The Journal FactCheck unit) takes into account the current publication frequency. In terms of the number of explainers published in the "FactFind" and "Explainer" sections, an average of 0.75 explanatory articles were published monthly on this platform in 2022.

Newtral typically summarises the consulted sources at the end of their explainers. However, two texts omitted this practice, and no accompanying explanation was provided for the omission. The Spanish outlet employed visual elements to enhance its explainers, appearing in 86 out the 149 pieces. These elements usually included screenshots of tweets, photographs, or images to illustrate the topic, graphics (often in-house-produced), and links to videos. Audience participation primarily occurred through comments on the articles, a feature that Newtral enabled for 87 of its 149 explainers. It is worth noting that the remaining 62 specific explainers, particularly those addressing topics related to anti-immigration and xenophobic, disinformation related to Catalan separatism, and COVID-19 vaccines, were not open to reader feedback.

In the case of Les Décodeurs, 48 explanatory texts were published in 2022, all within the "Pour Comprendre" section. These pieces were initially published once a week but saw an increase in frequency in 2023, typically with two explainers per week. The topics covered by Les Décodeurs' explanatory texts predominantly revolved around the French political context. However, they also featured international politics, focusing on Russia's invasion of Ukraine, as well as economic issues such as inflation and pension reform. Climate change and its impact on the economy were also addressed. Notably, COVID-19 received sparing coverage, with only one text identified in 2022, and subsequent pieces throughout the year did not revisit this topic. Other themes found, albeit to a lesser extent, included education, history and justice.

Regarding bylines, all the pieces are attributed to members of the Les Décodeurs team, except for four covering various topics (climate change, economics, national

politics and international politics). Additionally, one piece on justice is signed by *Le Monde*, the newspaper affiliated with this verification platform. Slightly over half of the published explainers, specifically 27, are individually signed, while the remaining pieces are attributed to several authors. Typically, two authors are listed, but in some cases, up to four authors contribute to a single article.

The sources consulted in the explainers are consistently and transparently identified, mainly where graphics are included. Among the 48 articles published in 2022, 22 of them incorporated graphics. Interestingly, most of these elements were not internally generated by the Les Décodeurs team, with only three being attributed to the team itself. Various sources are credited for these graphics, with a notable emphasis on using official sources specialising in statistics. User participation was practically non-existent, as no identified tool or method enabled this type of interaction within the pieces.

Throughout 2022, FACTA.news published a total of 82 articles that can be categorised as explanatory in nature. These pieces are distributed across two sections: "Articoli", created on 31 December 2021, featuring 55 articles, and "Storie", a section introduced in 2020, containing 27 such texts. The average frequency of publication was 1.5 articles per week. The pieces encompass a diverse range of topics, with the COVID-19 pandemic and Russia's invasion of Ukraine emerging as the overriding themes. Additionally, these explanatory pieces delve into subjects related to Italian and European politics, although this is a less frequently addressed area, as Pagella Política, a media outlet affiliated with FACTA.news and specialising in politics, primarily covers this. Other themes covered included science, climate change, sports and social networks. Once a month, FACTA.news publishes a piece summarising the topics that have generated the most disinformation, as determined by the Italian Digital Media Observatory. A similar article at the European level is produced, using the European Digital Media Observatory as its reference.

These articles aim to provide an in-depth exploration and context on topics that recurrently serve as sources of disinformation in Italy and Europe. It is worth noting that the "Articoli" section aims to include more complex articles with more context than those designed to alert readers to disinformation published in the "Antibufale" section. Conversely, "Storie" features more specialised, lengthier and in-depth pieces. In both instances, these can be categorised as explanatory texts. As such, all of them (excluding articles that provide summaries of the digital observatories' reports) conclude with a section highlighting the key takeaways.

With few exceptions, the explainers featured in FACTA.news are not attributed to specific journalists. Only 3 of the 82 articles are credited to a journalist. However, the sources referenced in the pieces are consistently and explicitly identified. As a general practice, the explainers include numerous links to other news items, websites, archives, and other resources, allowing readers to access the sources upon which the pieces are based. Out of the 82 published articles, slightly over half (44) incorporate visual elements to complement the explanations, primarily graphs, photographs, tables and embedded tweets. Notably, all articles published in the two sections maintain an open comments section and provide options for sharing the content via WhatsApp, Facebook, X (formerly known as Twitter), LinkedIn, Telegram and email. Most of the articles feature user feedback, usually one or two comments, although some of the more contentious pieces (concerning Italian politics related to xenophobia) generated up to 22 comments. In all cases, each comment is responded to by an administrator from FACTA.news.

Regarding The Journal FactCheck unit, it is evident that the platform published only nine explanatory texts in 2022 under the "FactFind" and "Explainer" formats, resulting in a frequency of 0.75 articles per month. The platform prioritised topics related to climate change and national politics, although it also addressed transport, education and tourism. Notably, in 2023, the platform increased the publication of this type of content and expanded the range of topics covered to encompass the treatment of social inequalities and health. As Shane Raymond points out, the selected topics "do not have to be Irish, but do need to have a relevance to our readers, an Irish connection".

Each explainer produced by The Journal FactCheck unit is individually crafted and bears an author byline. During the first part of 2022, several journalists from The Journal (Orla Dwyer, Michelle Hennessy, Stephen McDermott, Brianna Parkins and Céimin Burke) contributed explainers. Starting in September 2022, when Shane Raymond joined *The Journal*, most of the explanatory content had been authored by Shane Raymond and Stephen McDermott. On the rare occasions when part of the content was prepared by another organisation, such as Agence France-Presse, it was specified at the end of the text.

In each article featured in the "FactFind" and "Explainer" sections, the sources used are clearly referenced through hyperlinks, facilitating user access to the materials used. In particular, for questions such as "How many current TDs own houses that they rent out?" [45], an additional effort is made to ensure informational transparency by outlining the methodology used and the range of sources consulted in creating the article.

The Journal FactCheck unit routinely integrates multimedia elements into its texts, including illustrative images of the topics covered and screenshots of content published on X (formerly known as Twitter). The platform also occasionally includes tables and maps to visually represent numerical and graphical information, as seen in the explainers "Is the carbon tax on home heating oil an excise duty—and could it be cut?" [46] and "Is Navan the largest town in the country without a rail line?" [47].

At the end of each article, The Journal FactCheck unit reminds users that they can participate in the verification process by suggesting topics through the channels specified in the "Reader's Guide". Each explainer allows the audience to comment in the "Your voice" section. However, users do not consistently take part in this participation opportunity. Despite each explainer published in 2022 being viewed by an average of 28,300 readers, only three out of nine explainers received comments. The piece "Will planned turf regulations reduce air pollution?" [48] received the most feedback, with 52 comments.

### 4.2. Origin and Causes of Explanatory Journalism within Fact Checking and Working Methodologies (RQ2)

Newtral, The Journal FactCheck unit and FACTA.news each have two distinct sections where they publish explanatory articles. In the initially created section called "You ask us about"—which Joaquín Ortega describes as primarily explanatory—Newtral compiles topics sent in by users. These topics frequently require contextualisation when addressed due to the questions they generate. In contrast, the section "We explain it to you" serves as a "hook"—and also a catch-all, not exclusively for explainers—where readers can delve into subjects that the editorial team deems as being in need of more in-depth coverage. Ortega says the former corresponds to a reactive way of combatting disinformation, while the latter is proactive.

Meanwhile, The Journal FactCheck unit has a "FactFind" section for topics that "when even after looking into it we might not know the answer", and the "Explainer" section, which "is more for stuff where information is already there, but we add more context to the news" (Shane Raymond). FACTA.news's explanatory content is published in the "Articoli" section, whose goal is to "feature articles more that are complex than the simple debunkings included in the "Antibufale" section" (Giovanni Zagni), as well as in the "Storie" section, which is given over to even more specialised topics resulting from "original research" by the editorial staff, such as scientific themes related to the COVID-19 pandemic.

Newtral defines explainers as texts that do not have a structured format as traditional fact check reports do—in which there is "a presentation of evidence that leads you to a conclusion" (Joaquín Ortega)—and it claims that their aim is not so much to debunk as to explain a complex subject by contextualising it and including an expert's point of view. In essence, there is "more freedom in reporting". For Les Décodeurs, it has to do with "starting from a current event and going back to the past to find the explanation" (Jonathan Parienté). They also describe the format as "a luxury" because, for once, the journalist allows themselves the space and time needed.

### 4.2.1. Origin of Explainers: Causes and Objectives

Newtral began publishing explainers in "You ask us about" after launching its WhatsApp service in 2018. Through this channel, they began receiving verification requests from the audience. Much of this content cannot be classified as traditional hoaxes or disinformation by users, prompting the Newtral team to "explain where that headline came from, what the context was, what data might have been omitted", according to Joaquín Ortega.

FACTA.news, for its part, uses explainers to address "trends" and "narratives" of disinformation that the team detects during fact checking. They also use explainers in cases where traditional fact checking is not possible and, by extension, a final rating on the truthfulness or falsehood of the content cannot be determined. The "Articoli" section came into being after the "Storie" section, when the team realised that they needed a "more agile and flexible" format.

Les Décodeurs acknowledges doing less "pure fact checking" over time, as they find it less effective: "Fact checking is of little interest if we want to reach people we have to go to the explainer" (Jonathan Parienté).

The team at The Journal FactCheck unit frames explainers within the context of a fundamental need for citizens to "understand the world they live in" and believes that the format should give readers all the essential information to answer their inquiries independently.

When Newtral began fact checking, they took North American outlets as a reference, but it asserts that it did not look at other initiatives for explainers. In the case of Les Décodeurs, Parienté recollects that the project had small beginnings and lacked references, which he attributes to the project's success. It emerged in close contact with the readership.

### 4.2.2. Primary Themes and the Role of COVID-19

All the four outlets interviewed acknowledge that there are no strict rules governing their topic selection and that they cover a wide array of subjects in which current affairs and trends heavily influence their choices. In its day-to-day operations, the French outlet Les Décodeurs organises itself similarly to any other section of *Le Monde*, with editorial meetings considering the newspaper's established topic "hierarchy". However, it sometimes collaborates with specialists who suggest new subjects. During the coronavirus pandemic, a substantial portion of the platforms' content revolved around this issue:

> "It was an all-encompassing topic (...) from which you could generate all sorts of information" (Jonathan Parienté).

> "It was our duty to inform people and often this involved explaining things in a meticulous way, citing papers and quoting scientists" (Shane Raymond).

> "There were many things that journalists themselves did not understand, and the best way to explain these to people was to be able to understand them (...). From the perspective of disinformation dissemination, there was indeed a before and after" (Joaquín Ortega).

The pandemic marked the inception of FACTA.news in April 2022, since the initial project, Pagella Política, focused exclusively on political fact checking. Giovanni Zagni explains, "We felt the need for a new outlet focused on debunking disinformation on a broader scope". He adds, "COVID-19 set the stage for the entire launch of our debunking project, and explanatory articles were part of it from the beginning".

However, the pandemic alone does not account for the expansion of this new format, which had been deemed necessary for some time. According to Ortega, "The need to explain the noise and try to dispel it arose immediately after starting fact checking (...) and that came about before COVID", pointing to hyperconnectivity and information overload as motivational factors for the need to explain certain topics.

*4.3. Publication Frequency and Working Methodologies (RQ3)*

Generally, the platforms generally do not establish fixed publication frequencies, although they tend to maintain a more or less consistent rhythm. Currently, The Journal FactCheck unit produces a minimum of two articles per week, with one in the "FactFind" section, while Newtral publishes at least one daily. FACTA.news averages about one explainer per week.

The Irish platform indicates that it does not impose a specific timeframe for preparing explainers. Some topics require days, while others take weeks—similar to the approach at Les Décodeurs—and sometimes, more urgent matters arise during the process that must be prioritised.

Shane Raymond (The Journal FactCheck unit) adds that once the texts are completed, they undergo review by two editors before publication. FACTA.news follows the same procedure, with one of the editors having a senior-level profile. At Newtral, the pieces are reviewed by three people, although they do not perform as much data "double-checking" as in traditional verification because experts' opinions, for example, are not fact checked. As for Les Décodeurs, the general rule is to work in teams of three people.

Most of the sources used in explainers are official. The Journal FactCheck unit specifically mentions "research, statistics, papers, and government answers". Newtral follows a methodology very similar to traditional fact checking and maintains its policy of including all consulted sources at the end of the piece. However, in explainers, it does include quoted opinions from specialists to provide context. Les Décodeurs employs a "not overly normative" cross-referencing methodology for sources and asserts that it maintains distance from official sources when fact checking them. FACTA.news acknowledges following a methodology very similar to fact checking, but in the "Articoli" and "Storie" sections, the outlet adds "practical tips on how to spot disinformation and debunk dubious content" (Giovanni Zagni).

Regarding format, the Italian outlet publishes explainers of varying lengths, from one-page pieces (300–400 words) to longer articles (around 1000 words). While a "classic" fact checking article in The Journal Fact Check unit can be around 1000 words, "FactFind" pieces can extend to 4000 or 5000 words, although, as Shane Raymond notes, "there is no fixed criteria". Newtral aims to keep the texts to about a page and a half to prevent overwhelming the reader with so much information that they wind up "not understanding anything". Following this principle, the Spanish platform is experimenting with a new section called "In a graph", which seeks to explain a topic using a single graph or chart to provide a sense of its "magnitude". Les Décodeurs prioritises creating pieces with a simple structure, allowing for a structure of "a lot of information without confusing the reader".

In general, the explainers are written by in-house fact checkers. However, in the case of FACTA.news, some external contributors also occasionally write these pieces. Les Décodeurs divides tasks among teams based on the subject and required expertise.

The articles are typically signed by their authors. For Joaquín Ortega (Newtral), the practice relates to the necessary transparency they demand from others—they inform readers to be wary of unsigned articles—and to emphasis the journalist's responsibility for the information presented. FACTA.news is the only platform that follows a "non-signature" policy, although, as Giovanni Zagni acknowledges, this has evolved, and the final decision rests with the author of the text. Consequently, approximately half of the pieces are signed.

Jonathan Parienté highlights the value of having multidisciplinary profiles within the team, which allows Les Décodeurs to diversify its activities, including data journalism and international investigations: "(...) They are engineers and journalists, cartographers and journalists, etc., and this makes us self-sufficient in the production of articles and research".

Target Audience and Participation

Shane Raymond (The Journal FactCheck unit) emphasises that the goal is to reach as many people as possible, although their typical reader is currently under 40 years of age and non-specialised. From Newtral's perspective, they admit that they have never specifically defined their target audience, but they have realised that their WhatsApp service "is entirely cross-sectoral: for both right and left individuals, older people, young people alike (...)". Les Décodeurs is clear about targeting a different type of reader than the classic *Le Monde* profile (which is predominantly male and of a higher socio-professional status). FACTA.news does not pursue a specific audience profile but works to reach new audiences. In this regard, Giovanni Zagni believes that while the debunkings have a limited audience (as they typically arise from reader requests), articles published in the "Articoli" and "Storie" sections can reach new audiences.

Regarding the feedback audience participation, Les Décodeurs currently limits interactions to discussions with readers concerning complex topics. The outlet also tends to steer clear of platforms such as X (formerly known as Twitter), a stance also echoed by The Journal FactCheck unit, which notes that most responses on this social network are largely negative, although it does receive positive comments and emails from referencing its work. The Irish outlet acknowledges that the comments featured under fact checks "could be filled up with disinformation". Nevertheless, they have developed "three or four stories" based on reader requests via email. As Shane Raymond points out, "People have just put in suggestions that have been really helpful". Newtral describes its interaction with the audience as "journalism à la carte", more so in the case of fact checks than explainers, where the conclusion is unclear and therefore "you're not necessarily agreeing with anyone". At FACTA.news, they do not explicitly receive requests for explainers because they do not label explanatory texts as such. Still, they do receive numerous alerts and requests regarding complex topics that cannot be adequately addressed with a "simple debunking".

### 4.4. *The Role of Explanatory Journalism as an Ally in Combatting Disinformation (RQ4)*

The interviewees offered diverse perspectives on the importance of explanatory journalism in the fight against disinformation. Jonathan Parienté (Les Décodeurs) believes that, along with media literacy, explainers are the most effective tool to combat the problem, even surpassing traditional fact checking. For Shane Raymond (The Journal FactCheck unit), explanatory journalism has a bright future ahead, particularly since "as long as statistics are around, we are going to need explanatory pieces".

Giovanni Zagni (FACTA.news) believes that this type of journalism is "crucial in disentangling complex issues and helping people understand difficult debates". Therefore, explanatory pieces "can be useful to better inform the public and reduce the amount of noise and confusion, thus indirectly helping combat disinformation".

Newtral believes that within the current "deluge" of disinformation, assessing the true utility of explainers is more challenging because, unlike traditional fact checking, "it is not binary". Joaquín Ortega adds that this format may help people understand why certain topics hold importance, although he emphasises that these texts are not intended to "tackle clear and specific disinformation". Ortega also reflects on the possibility that the rating within traditional fact checks is showing signs of wear and tear:

> "It's very likely that many people have realised that you can't do pure and hard fact checking all the time because not everything can fit into a label".

In addition, he notes that at Newtral, explainers were understood from the outset as a type of content that could coexist and run in parallel with fact checking.

Giovanni Zagni (FACTA.news) believes that explanatory journalism cannot be considered fact checking per se, but he highlights the fact that it is increasingly present on fact checking platforms:

"The reason is probably found in the need to find new and broader audiences, as well as, more conceptually, contributing to a very audience-led, public service idea of journalism, that is at the core of many fact checking outlets".

## 5. Discussion and Conclusions

In today's communication ecosystem, which is characterised by multiple challenges and provocations, fact checking platforms have emerged as essential tools to combat the disinformation that threatens democratic well-being [34,49]. Despite sharing various essential characteristics, fact checking practices are not standardised. Verifiers are deploying an increasingly diverse array of formats to tackle disinformation. Beyond political fact checking, which laid the foundation for many verification projects [1,24], fact checking platforms are currently debunking ever more online disinformation [50]. However, this trend complements the adoption of other formats, such as explanatory journalism, that aim to provide context, depth and detail. In this constantly evolving landscape, this study has examined the adoption and application of the explainer format by leading fact checking platforms in various European countries, including Spain, France, Ireland and Italy.

Regarding the first research question (RQ1), the study reveals that this type of content still has a limited presence on fact checking platforms' websites compared with traditional fact check reports, and it does not have fixed schedules or lengths. In general, explainers are characterised as addressing a wide range of topics related to national and international current events. Despite this shared trait, there are divergent characteristics among the platforms studied. For instance, while Les Décodeurs typically features pieces crafted by multiple fact checkers, joint authorship is also common in Newtral, albeit less prevalent. The Journal FactCheck unit produces texts individually. In some cases, such as for Newtral, the sources used are listed at the end of the explainers, while in other cases, the sources are referenced within the body of the articles.

In response to the second and third research questions (RQ2, RQ3), it is concluded that three essential factors account for the rise of explanatory journalism within fact checking organisations. Firstly, the interviewed professionals emphasise the desire to transcend the limitations of traditional fact checking. Just as explanatory journalism is often defined in contrast to conventional journalism based on its goals, scheduling, length and format [12,13,17], verification platforms frequently draw analogies and distinctions between explainers and traditional fact checks. Explainers have a format closer to reportage and are less rigid than "conventional" fact check reports. They give professionals more freedom, flexibility and space to address topics and provide context without needing a final rating or label. In this sense, the explanatory format upholds the values of thoroughness and transparency [44] and reinforces the notion that verifiers serve as agents of context [51].

Secondly, through explainers, there is a clear intention to generate impact and interest among the citizens and to empower them to understand current affairs and make informed decisions. Thirdly, there is a desire to broaden the range of topics covered, incorporating the concerns and suggestions of readers into the agenda. This audience-orientated approach directly aligns with the concept of media accountability [52] and its importance in countering the erosion of public trust in information and its professionals [37,53].

The motivations articulated by the interviewees are closely related to the working methodologies they employ to create explainers, which are based on editorial rigour and the use of multiple filters, the absence of time constraints for content production, extensive reliance on data provided by official sources and experts and experimentation with visual elements to convey information clearly.

Finally, regarding the fourth research question (RQ4), the interviewees concur that the explainer is a highly valuable tool for combatting disinformation and educating the public. The explainer occupies a distinctive space within the information menu provided by fact

checkers, serving as complementary content to conventional fact checking reports. It is observed that the explanatory format is a rising trend with a promising future.

Overall, this study has allowed us to examine the adoption and use of explainer format by verification platforms in different communicative contexts. The results of this qualitative research should be seen in the light of their limitations. Each of the sampled organisations has its "unique features" [39] (p. 60) in terms of organisational structure, resources, verification practices, and approaches to explanatory journalism. Therefore, the results cannot be generalised to all fact checking organisations operating in the sampled countries (Spain, France, Italy and Ireland). While it is limited to four verification projects in Europe, this contribution can assist other fact checkers representing diverse editorial models and cultures to understand the characteristics, methodologies and strategic applications of this format. Furthermore, this study can heighten awareness of such pieces among users, thus enabling new opportunities for citizen participation in fact checking consumption and practice. In the academic sphere, this research aspires not only to benefit the training of future fact checkers but also to motivate the development of new research in various directions.

Subsequent studies should monitor the future use of the explanatory format by fact checkers in different regions and interview their managers to further unravel their professional practices, methodologies, resources and specific challenges. From a production standpoint, it is imperative to explore how verification platforms can establish national and transnational collaborations to produce explainers of exceptional complexity. Through reception techniques such as focus groups and interviews, it will be pertinent to scrutinise how explanatory pieces are consumed by the audience, which topics are prioritised by them, and what drives them to propose specific treatments. These approaches will contribute to attaining a comprehensive and multifaceted understanding of the evolution of explanatory journalism within the field of fact checking.

**Author Contributions:** Conceptualization, V.M.-G.,X.R.-V., R.R.-M. and M.M.-R.; methodology, V.M.-G., X.R.-V., R.R.-M. and M.M.-R.; software, V.M.-G., X.R.-V., R.R.-M. and M.M.-R.; validation, V.M.-G. and X.R.-V.; formal analysis, V.M.-G., X.R.-V., R.R.-M. and M.M.-R.; investigation, V.M.-G., X.R.-V., R.R.-M. and M.M.-R.; resources, V.M.-G., X.R.-V., R.R.-M. and M.M.-R.; data curation, V.M.-G.; writing–original draft preparation, V.M.-G., X.R.-V. and R.R.-M.; writing–review and editing, V.M.-G., X.R.-V., R.R.-M. and M.M.-R.; visualization, V.M.-G.; supervision, V.M.-G. and X.R.-V.; project administration, V.M.-G.; funding acquisition, R.R.-M. and M.M.-R. All authors have read and agreed to the published version of the manuscript.

**Funding:** This study is part of the research project "Media Accountability Instruments against Disinformation: The Impact of Fact Checking Platforms as Media Accountability Tools and Curricular Proposal," funded by the Spanish Ministry of Science and Innovation and the State Research Agency (FACCTMedia, PID2019—106367GB I00/AEI/10.13039/501100011033).

**Informed Consent Statement:** Informed consent was obtained from all subjects involved in the study.

**Data Availability Statement:** The data presented in this study are available in Tables 1 and 2.

**Conflicts of Interest:** The authors declare no conflict of interest.

## Appendix A  Interview Scripts

1. What does your platform define as explanatory journalism?
2. Since when have you been producing explanatory articles? What led to the decision to adopt this format, and where did the need for it arise? Were you inspired by any other fact checking platform or media outlet?
3. What topics do your explainers focus on? How are the themes chosen for coverage? Do you specialise in any particular areas?
4. Has the COVID-19 pandemic played a role in promoting this type of format?

5.  How frequently do you publish explanatory articles? Do they have a set publication schedule? What is the typical length of these articles?
6.  What methodology do you employ to create explainers? What steps do you follow, types of sources do you consult, and number of review layers do you apply before publication? Do explainers sometimes emerge from previous fact checks?
7.  Is this type of content authored by your regular team members or external contributors, and is the journalist's byline typically included?
8.  How does this methodology differ from the procedures you use to develop 'traditional' fact checks?
9.  Is there a specific target audience for these explanatory pieces?
10. What kind of feedback do you receive from your audience regarding these explanatory articles, and do users request explainers on specific topics?
11. In your opinion, how can explanatory journalism aid in combatting disinformation, particularly in dealing with complex issues?
12. Do you believe that explanatory journalism currently holds enough significance to be considered a new trend in fact checking? If so, why? If not, why not?

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
