# Peer review of "Explanatory Journalism within European Fact Checking Platforms: An Ally against Disinformation in the Post-COVID-19 Era"

_societies, doi:10.3390/soc13110237_

Round 1

Reviewer 1 Report

Comments and Suggestions for Authors

The paper deals with a contemporary, fascinating topic, which is important for today's reality.  The authors did a great job delving into the subject, and I really enjoyed reading it!

This study investigates 7 the characteristics and methodologies of contemporary explanatory journalism by analysing four European verification platforms (Newtral in Spain, Les Décodeurs in France, FACTA.news in Italy and The Journal FactCheck unit in Ireland). 

The authors employed a content analysis of a corpus of explainers and semi-structured interviews with the managers of these outlets. Findings reveal that explainers encompass a wide range of topics, typically revolving around current affairs. These pieces are usually authored by fact-checkers and published, with bylines, within dedicated sections that encourage audience participation. Explainers do not adhere to a fixed periodicity or length and adopt a format similar to feature articles, displaying a degree of flexibility. They leverage data provided by experts, official sources, and employ visual elements to convey information clearly. The interviewed managers concur that explanatory journalism represents an invaluable tool in combatting disinformation and has a promising future ahead. 

On page 4, it is noted that explanatory texts published on the selected platforms during 2022 were examined, and interviews were conducted via videoconference between November 2022 and March 2023 and were recorded for subsequent transcription and qualitative analysis. Were the explanatory texts published and collected all through the year? or also during the same months as the interviews? it wasn't clear. 

Author Response

October 27, 2023

Dear editor, 

Please, find attached the last version of our manuscript “Explanatory journalism within European fact-checking platforms: an ally against disinformation in the post-COVID-19 era”. We really appreciate your time and the comments and suggestions made by the reviewers, which we consider very appropriate and that clearly contribute to improve the text.

In the following table, we respond to the reviewers' comments and explain the changes we have made. All of them have been highlighted in red in the manuscript.

Reviewer 1 Comments and suggestions

Author’s answer

The paper deals with a contemporary, fascinating topic, which is important for today's reality.  The authors did a great job delving into the subject, and I really enjoyed reading it!

This study investigates 7 the characteristics and methodologies of contemporary explanatory journalism by analysing four European verification platforms (Newtral in Spain, Les Décodeurs in France, FACTA.news in Italy and The Journal FactCheck unit in Ireland).

The authors employed a content analysis of a corpus of explainers and semi-structured interviews with the managers of these outlets. Findings reveal that explainers encompass a wide range of topics, typically revolving around current affairs. These pieces are usually authored by fact-checkers and published, with bylines, within dedicated sections that encourage audience participation. Explainers do not adhere to a fixed periodicity or length and adopt a format similar to feature articles, displaying a degree of flexibility. They leverage data provided by experts, official sources, and employ visual elements to convey information clearly. The interviewed managers concur that explanatory journalism represents an invaluable tool in combatting disinformation and has a promising future ahead.

On page 4, it is noted that explanatory texts published on the selected platforms during 2022 were examined, and interviews were conducted via videoconference between November 2022 and March 2023 and were recorded for subsequent transcription and qualitative analysis. Were the explanatory texts published and collected all through the year? or also during the same months as the interviews? it wasn't clear.

We really appreciate the positive evaluation regarding the main aspects of the paper. Thank you very much.

In line 189 we specify that we analyze the “explanatory texts published on the selected platforms during 2022”. However, we have included that information again in line 225, so that this aspect is absolutely clear for the reader. Thank you for your helpful comments.

Reviewer 2 Report

Comments and Suggestions for Authors

The text, in general, presents an appropriate structure, it is clear and very relevant to the current context. Additionally, there are interesting and well-expressed ideas in the conclusion section. However, in the introduction section, it is necessary to explain better the importance of fact-checkers in comparison to the role of combatting disinformation carried out by public service media. This would enrich the research context. In this regard, it is recommended to consult the following article for further information: Horowitz, M., Cushion, S., Dragomir, M., Gutiérrez Manjón, S., & Pantti, M. (2022). A framework for assessing the role of public service media organizations in countering disinformation. Digital journalism, 10(5), 843-865. https://doi.org/10.1080/21670811.2021.1987948

Comments on the Quality of English Language

 It is necessary to review some connectors and orthotypical errors in the text. In some paragraphs, there is more space than in others. It is important to revise this to improve the text's style. Additionally, to facilitate the reading of the results, it is recommended to shorten the length of paragraphs, as some of them exceed 10 lines without breaks in between.

Author Response

October 27, 2023

Dear editor, 

Please, find attached the last version of our manuscript “Explanatory journalism within European fact-checking platforms: an ally against disinformation in the post-COVID-19 era”. We really appreciate your time and the comments and suggestions made by the reviewers, which we consider very appropriate and that clearly contribute to improve the text.

In the following table, we respond to the reviewers' comments and explain the changes we have made. All of them have been highlighted in red in the manuscript.

Reviewer 2 Comments and suggestions

Author’s answer

The text, in general, presents an appropriate structure, it is clear and very relevant to the current context. Additionally, there are interesting and well-expressed ideas in the conclusion section. However, in the introduction section, it is necessary to explain better the importance of fact-checkers in comparison to the role of combatting disinformation carried out by public service media. This would enrich the research context. In this regard, it is recommended to consult the following article for further information: Horowitz, M., Cushion, S., Dragomir, M., Gutiérrez Manjón, S., & Pantti, M. (2022). A framework for assessing the role of public service media organizations in countering disinformation. Digital journalism, 10(5), 843-865. https://doi.org/10.1080/21670811.2021.1987948

We appreciate the positive feedback and suggestions made by the reviewer. However, we don’t consider it pertinent to delve into the importance of fact-checkers in comparison to the role of combatting disinformation carried out by public service media (PSM) since this would fall out the scope of the paper –which is focused specifically on explanatory journalism within fact-checking platforms (all of them private). Moreover, if we talked about PSM in relation to fact-checking, we should also extend the explanation to any forms of private media in the four countries where the selected platforms currently operate, and this would lead us to deviate from the main topic.

In any case, we thank the reviewer for the comments, and we have specified the following in lines 154 and 155: “we analysed four private fact-checking organisations”.

It is necessary to review some connectors and orthotypical errors in the text. In some paragraphs, there is more space than in others. It is important to revise this to improve the text's style. Additionally, to facilitate the reading of the results, it is recommended to shorten the length of paragraphs, as some of them exceed 10 lines without breaks in between.

Thank you for your suggestions. We have reviewed the manuscript and shorten the length of some paragraphs (lines 87, 109, 159, 214, 301 and 366).

We have also corrected some errors:

-In line 109 we added a full stop replacing the comma.

-In line 249 we added a full stop replacing the comma.

-In line we added 459 a full stop at the end of the sentence.

The text was translated by a native translator who is an experienced professional. This person has a wide experience in translating academic papers within this specific field of knowledge.

Reviewer 3 Report

Comments and Suggestions for Authors

Interesting approach to explanatory journalism practiced by four European fact-checkers. Despite the international focus, the following weaknesses are identified.

1) Some of the research questions should be separated independently, for example, RQ2.

2) Some of the questions to the interviewees reiterate the results obtained in the content analysis. 

3) From a methodological standpoint, these are four case studies, and their practices cannot be generalized to the entire country. (No limitations of the study have been presented). More information about how the qualitative analysis was conducted is also needed.

In the qualitative section (based on interviews), many statements are presented, but the researcher does not analyze them; they are transcribed as if the researcher were a stenographer.

4) Regarding the results, before delving into quantitative results independently, the author should create an introduction that provides a collective systematization of the results, followed by individual perspectives. This would help avoid a disjointed approach.

5) In lines 107-109, I suggest mentioning each geographic location only once.

6) The article requires revision by the authors. Some editing errors have been detected: line 109: "Latin America, [36] At the same time"; line 129 "the El Explicador", Line 242: "explainers, It is "...

7) In line 155, specify the date on which Les Décodeurs' status has expired to avoid misunderstandings.

8) Finally, I would suggest to include this reference:

Viorela Dan & Doreen Rauter (2023) Explanatory Reporting in Video Format: Contrasting Perceptions to Those of Conventional News, Journalism Practice, 17:5, 1046-1067, DOI: 10.1080/17512786.2021.1966644 

Comments on the Quality of English Language

The quality of the English is acceptable, although we can find some punctuation errors and word repetitions.

Author Response

October 27, 2023

Dear editor, 

Please, find attached the last version of our manuscript “Explanatory journalism within European fact-checking platforms: an ally against disinformation in the post-COVID-19 era”. We really appreciate your time and the comments and suggestions made by the reviewers, which we consider very appropriate and that clearly contribute to improve the text.

In the following table, we respond to the reviewers' comments and explain the changes we have made. All of them have been highlighted in red in the manuscript.

Reviewer 3 Comments and suggestions

Author’s answer

Interesting approach to explanatory journalism practiced by four European fact-checkers. Despite the international focus, the following weaknesses are identified.

1) Some of the research questions should be separated independently, for example, RQ2.

Thank you for your helpful suggestion. We have separated RQ2 in two different research questions and made the necessary changes within the result section so there is a correspondence between the points 3 (Objectives and methodology), 4 (Results) and 5 (Conclusions) of our manuscript.

2) Some of the questions to the interviewees reiterate the results obtained in the content analysis. 

We appreciate the comment. During the research, we employed a mixed-method approach for that very reason, so that the results of both analyses could validate each other.

3) From a methodological standpoint, these are four case studies, and their practices cannot be generalized to the entire country. (No limitations of the study have been presented). More information about how the qualitative analysis was conducted is also needed.

We have added the following sentence and reference within the ‘Objectives and methodology’ section to reinforce the value of triangulation as a technique that allows researchers to corroborate data and, furthermore, to get a deeper insight on the topic under investigation:

“To respond to these questions, we employed a mixed-method approach. Bringing together quantitative and qualitative techniques allowed researchers to “triangulate findings in order to mutually corroborate them” [39] (p. 557) while helping them to obtain a fuller account of the phenomena under study.”

This reference has been added at the end of the manuscript:

Clark, T.; Foster, L.; Sloan; Bryman, A. Bryman’s Social Research Methods. Oxford University Press, Oxford, UK, 2021.

In the qualitative section (based on interviews), many statements are presented, but the researcher does not analyze them; they are transcribed as if the researcher were a stenographer.

The statements are presented as part of the results obtained through the interviews and, with all this valuable material, we later write the discussion and conclusions in the final section.

4) Regarding the results, before delving into quantitative results independently, the author should create an introduction that provides a collective systematization of the results, followed by individual perspectives. This would help avoid a disjointed approach.

We appreciate the observation of Reviewer 3 on this matter. However, we consider that the main results of the paper are already presented in the abstract and later discussed in section 5 (Discussion and conclusions) in the light of the research questions that guided the study. From our perspective, including a summary of the findings at the beginning of section 4 would be redundant with such information. 

5) In lines 107-109, I suggest mentioning each geographic location only once.

We believe that it is very important to specify the countries each of the mentioned studies focus on, even when that means repeating some geographic locations twice or more times.

6) The article requires revision by the authors. Some editing errors have been detected: line 109: "Latin America, [36] At the same time"; line 129 "the El Explicador", Line 242: "explainers, It is "...

Thank you for the helpful suggestion. We have already corrected those mistakes and the changes are highlighted in red in the last version of the manuscript.

7) In line 155, specify the date on which Les Décodeurs' status has expired to avoid misunderstandings.

We have added the expiration date (August 20, 2021).

8) Finally, I would suggest to include this reference:

Viorela Dan & Doreen Rauter (2023) Explanatory Reporting in Video Format: Contrasting Perceptions to Those of Conventional News, Journalism Practice, 17:5, 1046-1067, DOI: 10.1080/17512786.2021.1966644 

Thank you for your suggestion. This reference was already included in our original manuscript (number 13). This reference was used to provide a proper explanation of how explanatory journalism can facilitate a greater understanding of complex social issues and thus help contribute to an informed citizenry.

The quality of the English is acceptable, although we can find some punctuation errors and word repetitions.

We appreciate your comments and we have reviewed the manuscript. Thank you.

*We have also included a bracket, corrected the figure 1,4 (1.4) and the word “LGBTI” twice (both changes in table 2).

Round 2

Reviewer 3 Report

Comments and Suggestions for Authors

I appreciate the improvements applied by the authors. However, two relevant recommendations were not accepted. 1) No limitations of the research have been included; 2) The research is focused on four case studies, their practices cannot be generalized to the entire country. This approach should be explained as a limitation, also the selection criteria. 

Author Response

Dear reviewer,

Thank you once again for your helpful comments. We have responded to them by adding the following parts in the last version of the manuscript:

-Lines 166-171:

On the other hand, Newtral falls within for-profit organisations, as it began as a startup that, in addition to fact-checking, engages in other activities in audiovisual production. The four organizations have pioneered in introducing and consolidating the use of explanatory articles within their fact-checking propositions. As with other qualitative research, the study endeavored to understand the practices, methodologies and strategies applied by those organizations rather than aiming to generalize the findings to all the population [39], that is, to all the fact-checking platforms operating in the four countries. 

-Lines 595-600:

Overall, this study has allowed us to examine the adoption and use of explainer format by verification platforms in different communicative contexts. The results of this qualitative research should be seen in the light of their limitations. Each of the sampled organizations has its “unique features” [39] (p. 60) in terms of organizational structures, resource, verification practices, and approaches to explanatory journalism. Therefore, results cannot be generalized to all fact-checking organizations operating in the sampled countries (Spain, France, Italy, and Ireland). While it is limited to four verification projects in Europe, this contribution can assist other fact-checkers…

The authors